# Uptake and perceptions of oral HIV self-testing delivered by village health teams among men in Central Uganda: A concurrent parallel mixed methods analysis

Joanita Nangendo[1,2]*, Anne R. Katahoire[1], Charles A. Karamagi[1], Gloria O. Obeng-Amoako[1,3], Mercy Muwema[1], Jaffer Okiring[1,2], Jane Kabami[1,2], Fred C. Semitala[1,2,4], Joan N. Kalyango[1], Rhoda K. Wanyenze[5], Moses R. Kamya[1,2]

1 School of Medicine, Makerere University College of Health Sciences, Kampala, Uganda, 2 Infectious Diseases Research Collaboration, Kampala, Uganda, 3 International Centre for Evaluation and Development, Tema, Ghana, 4 Makerere University Joint AIDS Program, Kampala, Uganda, 5 School of Public Health, Makerere University College of Health Sciences, Kampala, Uganda

* joannangendo@gmail.com

**Data Availability Statement:** Data can accessed by contacting the School of Medicine higher degrees

## Abstract

The World Health Organization (WHO) recommends HIV self-testing (HIVST) to increase access to and utilization of HIV services among underserved populations. We assessed the uptake and perceptions of oral HIVST delivered by Village Health Teams (VHTs) among men in a peri-urban district in Central Uganda. We used a concurrent parallel mixed methods study design and analyzed data from 1628 men enrolled in a prospective cohort in Mpigi district, Central Uganda between October 2018 and June 2019. VHTs distributed HIVST kits and linkage-to-care information leaflets to participants in 30 study villages allowing up-to 10 days each to self-test. At baseline, we collected data on participant socio-demographics, testing history and risk behavior for HIV. During follow-up, we measured HIVST uptake (using self-reports and proof of a used kit) and conducted in-depth interviews to explore participants' perceptions of using HIVST. We used descriptive statistics to analyze the quantitative data and a hybrid inductive, and deductive thematic analysis for the qualitative data and integrated the results at interpretation. The median age of men was 28 years, HIVST uptake was 96% (1564/1628), HIV positivity yield was 4% (63/1564) and reported disclosure of HIVST results to sexual partners and significant others was 75.6% (1183/1564). Men perceived HIVST as a quick, flexible, convenient, and more private form of testing; allowing disclosure of HIV test results to sexual partners, friends and family, and receiving social support. Others perceived it as an opportunity for knowing or re-confirming their sero-status and subsequent linkage or re-linkage to care and prevention. Utilizing VHT networks for community-based delivery of HIVST is effective in reaching men with HIV testing services. Men perceived HIVST as highly beneficial but needed more training on performing the test and the integrating post-test counseling support to optimize use of the test for diagnosing HIV.

research ethics committee at Makerere University College of Health Sciences via email (rresearch9@gmail.com).

**Funding:** The research reported in this publication was funded by the Fogarty International Center of the National Institutes of Health under Award Number D43 TW010037 to JN. The content is solely the responsibility of the authors and does not necessarily represent the official views of the National Institutes of Health. The funders had no role in study design, data collection and analysis, decision to publish, or preparation of the manuscript.

**Competing interests:** The authors have declared that no competing interests exist.

# Introduction

Approximately 85% of people living with HIV (PLHIV) globally knew their HIV status in 2021 [1]. However, in Sub-Saharan Africa (SSA), men still lag behind women in HIV status awareness at 79% versus 87% [2]. Engaging men in HIV prevention and care services is a priority for achieving the end AIDS 2030 goals [3, 4]. In Uganda, as of August 2022, 76.1% of men aged 15–64 years were aware of their HIV-positive status compared to 83.5% of women [5]. Hindrances to testing among men in Uganda and SSA have been documented [2, 6–10], and in part explain the prevailing risk of infection and slow progress on the HIV prevention and care cascades. Men are highly mobile and less available because of work, leisure and social engagements [11], so are often missed in population-wide HIV services [7, 12]. Men are reluctant to test because of fear of injections, receiving HIV-positive results [10], and the discomfort of seeking care in a female-oriented healthcare system [13, 14]. They also experience higher stigma than women [15, 16], and struggle to overcome masculinity norms that underscore health seeking [8, 9, 17].

HIV self-testing (HIVST) can increase access to HIV testing services in the underserved and priority sub-groups including men in the high- [18–20], middle-, and low- resource settings [21–30]. HIVST is approved in Uganda [31], and has been successfully implemented among female sex workers, fisher-folk, injecting drug users [32], and male partners of pregnant women [33]. For these populations, HIVST is provided at selected public health facilities, with the support of HIV implementing partners [31]. Increasing uptake of HIV testing services among men requires careful selection of models to deliver effective interventions to the intended users, and a good understanding of the relationships that influence utilization of health services [34]. In Uganda, men's understanding and perceptions of HIVST are still under-explored yet could inform the design and delivery of HIV testing services to them. Furthermore, village health teams (VHTs) have been central in the scaling-up of primary health care services including; immunization, malaria control, nutrition, sanitation-, hygiene- and community HIV testing campaigns, as well as establishing linkages between health providers and end-users [35, 36]. Leveraging VHTs could advance community-based delivery of HIVST to enable testing among individuals like men who are often missed by universal mobile and facility-based HIV services [31]. We utilized VHTs to deliver oral HIVST to men via their households and offered up to 10 days for each participant to self-test. We assessed the uptake and perceptions of oral HIVST among men in Mpigi district, Central Uganda.

# Methods

## Study design and setting

We conducted a concurrent parallel mixed methods study from November 2018 to June 2019 to assess the uptake and perceptions of VHT-delivered oral HIVST among men in Mpigi district, Central Uganda. This is a peri-urban settlement around Lake Victoria whose population predominantly engages in subsistence farming, retail trading of food, fish and local merchandise, and motor-cycle and taxi transport [37]. This region has a very high adult HIV prevalence of 8.1%, and a prevalence of 5.9% among men [5]. Based on the 2021 national population assessment, 76.2% of men in this region had ever-tested for HIV but only 40.6% had tested in the past 12 months [5]. From our earlier research in the same area, 16% of men in Mpigi district had never tested while 66% had tested in the past 12 months [10]. However, the district has numerous facilities under the government and private-not-for-profit schemes that provide essential health care including HIV services, supported by over 2500 VHTs [37, 38].

## Study participants

Participants were eligible for the cohort if; aged $\geq$ 15 years, residents of the sampled village for $\geq$ 3-months prior to the study and without intention of relocating in the next 6-months, and had access to or owned an active mobile phone contact for future communication. In Uganda, persons aged 12–17 years can consent for HIV-testing as mature and emancipated minors [39], and legally permitted to consent for research [40]. We excluded participants if: they could not speak English or Luganda (a local language spoken by the majority of the population in this region), or had tested for HIV in $\leq$ 3-months prior to the study as per the Uganda Ministry of Health (MoH) recommendations on targeted testing.

For the qualitative study, we purposively selected and invited participants who were willing to share their perceptions of HIVST from among those enrolled in the cohort.

## Sampling and sample size estimation

Details of the sampling strategy were previously reported in our preliminary work on men's HIV testing in the same area [10]. Briefly, we employed two-stage cluster sampling. First, we used the formula by Bennet et al (C = P(1-P)*D/S2b) [41] to estimate clusters (C). We considered 52% as the prevalence (P) of HIV testing among men in Central Uganda at that time [42], a design effect (D) of two (2), standard error for the alpha value at 95% confidence interval of 0.0255, number of households per cluster (b) of 30; for convenience, and factored for 10% non-response. We computed 29 clusters, then approximated to the 30*30 strategy of the WHO Expanded Program on Immunization [41]. Ultimately, we selected 30 villages using proportionate-to-size probability in which we systematically sampled 30 households and enrolled all eligible participants.

For the qualitative study, we purposively sampled participants who self-tested regardless of the results obtained. We aimed to achieve maximum variation for age, HIV testing history, education and residence, and determined the minimum required number of in-depth interviews by data saturation [43].

## Study procedures

The details of our study procedures are published elsewhere [44]. Briefly, one VHT from each of the 30 study villages (in total 30 VHTs) who had received prior training about the study led the team to: sample households, determine eligibility of household members and explain the research to the community and local authorities. The VHTs used colored test kit pictorial inserts to demonstrate how oral HIVST works.

At enrolment, each participant received; an oral HIVST kit, a disposal paper bag in which to place the kit after self-testing, and a linkage-to-care leaflet listing facilities in the district with HIV testing and treatment services. During the visit, research assistants (RAs) administered a structured questionnaire in face-to-face interviews to collect participants' data on; socio-demographics (age, religion, education, marital status, occupation and likely time of being at home), circumcision status, HIV risk behavior, and testing history (lifetime testing, testing in the past 12 months; as a proxy of recent testing [31], couple testing and HIV-status disclosure). We defined HIV risk behavior as a composite variable of engaging in any of the following behaviors; multiple sexual partners, inconsistent condom use with sexual partner(s), transactional sex, and use of alcohol or drugs.

In ten days after distributing HIVST kits, we re-interviewed the participants to check if they self-tested or not, and to capture perceptions of the oral HIVST process. Since HIVST was relatively a new testing approach, offering 10 days for HIVST was considered sufficient time for each participant to reflect and decide to test or not, comparable to studies on community-

based unsupervised oral HIVST held elsewhere in SSA [45]. We considered a self-report of using the HIVST kit within 10 days of receipt as a measure of uptake, and verified all self-reports by viewing the used kit. During the follow up survey, we also invited purposively selected participants for in-depth interviews to share their perceptions of HIVST. We held 14 audio-recorded in-depth interviews (six with those who had positive HIVST results, and eight with those that tested negative) lasting on average 30 minutes. Interviewers were male social workers with a minimum academic qualification of a diploma who were competent in qualitative data collection, and fluent in Luganda and English. They used a pre-set topic guide comprising of open-ended questions translated to both languages to suit the preference of each participant. All interviews were held at a mutually agreed time and place convenient to the participant, and were audio-recorded for future analysis after seeking permission of the respondent.

## Ethical considerations

Our study was approved by the School of Medicine research ethics committee at Makerere University College of Health Sciences; REC REF# 2017–136 and the Uganda National Council for Science and Technology; HS226ES. We also received administrative permission from the Ministry of Health directorate in Mpigi district and local area authorities in all study villages. All prospective participants gave full written consent prior to joining the study.

## Data analysis

We analyzed our quantitative data using descriptive statistics and expressed HIVST uptake as a percentage with 95% confidence intervals (CI) adjusted for clustering at the village level.

For the qualitative data, we followed the Consolidated Criteria for Reporting Qualitative Research (COREQ) guidelines. An independent RA transcribed interview audio recordings verbatim while another translated Luganda transcripts to English, which we ultimately analyzed. Prior to starting analysis, the study lead (JN) reviewed all transcripts for accuracy and completeness, by concurrently reading and listening to the audios. We used thematic analysis employing a hybrid inductive-deductive approach [46], in six-steps including familiarization, coding, generating, reviewing, defining and naming, and writing up themes from the data [47]. JN generated the initial open codes, grouped them into categories and themes, and shared the initial codebook with two colleagues (MM and GOO) for independent review then the revised version was sent to a senior qualitative researcher (ARK) for final consensus. Feedback shared by AK guided the refining of final code groups, themes and selection of illustrative quotes.

Throughout the analysis, reflections made on the data were guided by Aday and Andersen's framework on the relationships influencing utilization of health care services which comprises of five domains; health policy, characteristics of the health care delivery system, characteristics of the population at risk, utilization of health care services, and consumer satisfaction [34]. We grouped the resulting themes into positive and negative perceptions of HIVST, and suggestions for implementation. Finally, we integrated the results from the quantitative and qualitative data for interpretation and discussion of the study.

## Results

### Participant flow chart

We sampled 1295 households and enrolled 1628 men in 30 villages. Each participant received one oral HIVST kit to test for HIV. Twenty-five (25) participants did not use the HIVST kit,

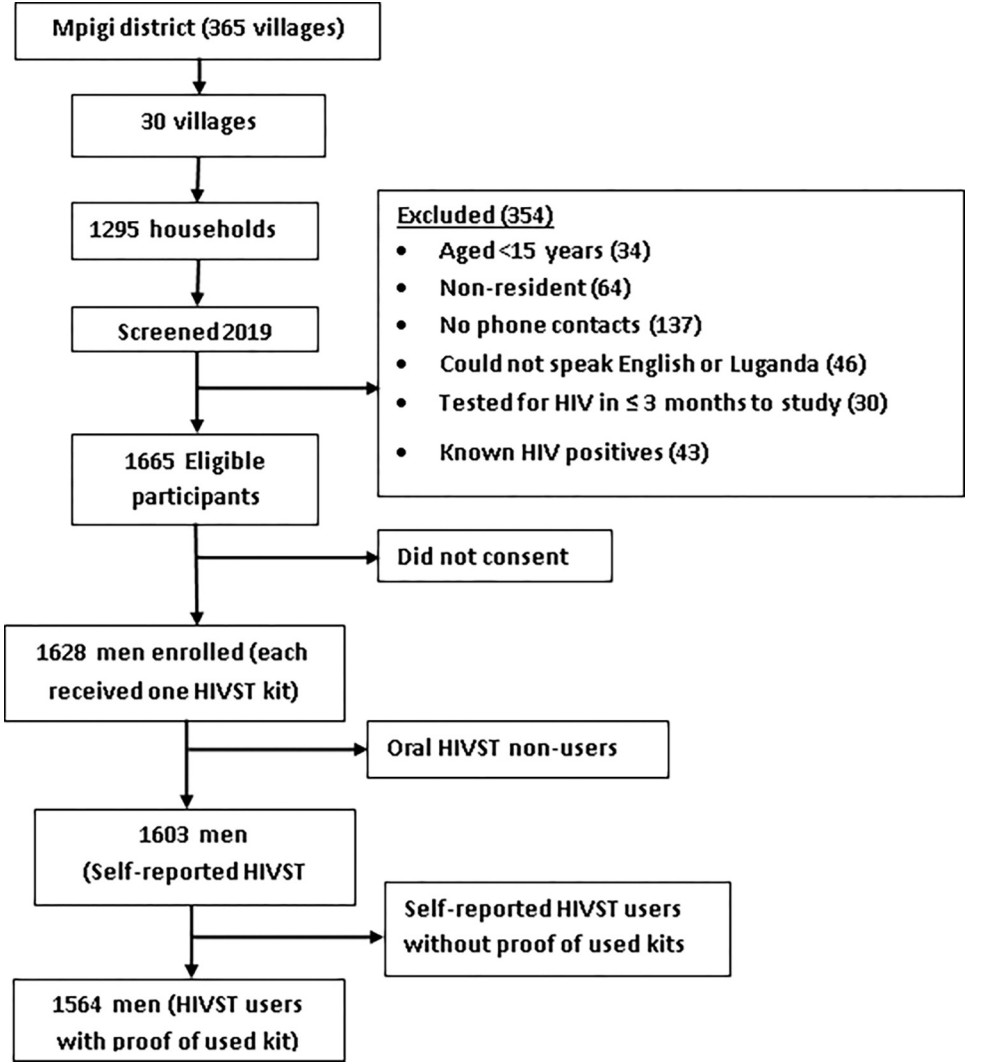

**Fig 1. Study profile showing participant enrolment in the oral HIVST cohort.**

while 39 of those who self-tested did not present proof of use (i.e. having a test stick from the used kit).

**Fig 1** shows the flow of participants in the study.

From among the participants in the quantitative study, we conducted in-depth interviews with 14 men who had self-tested, of whom eight (8) had tested HIV negative, and six (6) had tested HIV positive on HIVST.

## Socio-demographic and household characteristics

Participants' median age was 28 years; 81.8% were Christian; 54% had attained primary level education; 45% were involved in farming; and 74.9% of participants were currently married **Table 1** summarizes the participants' socio-demographic and household characteristics.

The majority of the participants were young of median age 28 years. They reported that HIVST provided quick and convenient testing for young people in cases of unplanned sex with casual partners.

**Table 1. Socio-demographic and household characteristics of participants enrolled in the oral HIV self-testing cohort study in Mpigi district, October 2018 (N = 1628).**

| Variable | Measure |
|---|---|
| Median age, (Interquartile range) | 28 (21, 38) |
| Religion, n (%) | |
| Muslim | 276 (17.0) |
| Christian[a] | 1333 (81.8) |
| Others* | 19 (1.2) |
| Education, n (%) | |
| No formal education | 82 (5.0) |
| Primary (P1-P7) | 879 (54.0) |
| Secondary & Tertiary | 667 (41.0) |
| Occupation, n (%) | |
| Farming | 733 (45.0) |
| Professional[+] | 269 (16.5) |
| Business/trading | 559 (34.3) |
| Unemployed & others[#] | 67 (4.2) |
| Marital status, n (%) | |
| Married | 1219 (74.9) |
| Unmarried [b] | 409 (25.1) |
| Time of day at home, n (%) | |
| 6:00am-12:00pm | 746 (45.8) |
| 12:01–6:00pm | 362 (22.2) |
| 6:01pm & beyond | 520 (32.0) |
| Median household size, (interquartile range) | 3 (1, 5) |
| Participant is a household head (Yes) n (%) | 1499 (92.1) |

[a] Christian include; Anglicans (279), Catholics (963), Pentecostals (91)

[b] Unmarried include; the single (304), and the widowed and divorced (105)

* Others include; Seventh day Adventist (15), Atheist (1) & Isa Masiya [Jesus the Messiah] (3)

[+] Professional are those practicing in a line of work after attaining formal qualification

[#] Students (11)

*"It is a good method because for us young people, where we go for parties you can get a girl speedily, that is where I can get this test from my bag and use it on her then decide to go on or not depending on the results." (Male, 32 years, HIV negative)*

Nearly three quarters of the participants reported that they were married, 30.6% had disclosed the intention to use HIVST to their partner, 31.3% asked their partner for help during HIVST, and 34.2% reported disclosure of the HIVST results to their sexual partners. Men reported that HIVST made it easy to disclose both their intention to test and HIV results to their partners and significant others for social support and guidance on the next steps.

*". . .I went and explained to her, tested and also took my results to her and I even taught her what I had learnt" (Male, 38 years, HIV negative)*

*". . .if they find out that I have HIV, I will also start medication like my friends . . .who were going to die but when they tested positive, they started medication." (Male 23 years, HIV negative)*

**Table 2. HIV testing history and behavioral characteristics of participants enrolled in the oral HIV self-testing cohort study in Mpigi district, 2018 (N = 1628).**

| Variable | Number (percentage) |
|---|---|
| Circumcised (Yes), n (%) | 797 (49.0) |
| HIV risky behavior* (Yes), n (%) | 1606 (98.7) |
| HIV testing history, n (%) | |
| Never tested | 322 (19.8) |
| Tested more than 12 months ago | 700 (43.0) |
| Tested in last 12 months | 606 (37.2) |
| Aware of HIV services in the community (Yes), n (%) | 1513 (92.9) |
| Testing for HIV at a community service point (Yes)[#], n (%) | 1057 (80.9) |
| Community testing service used[+] n (%) | |
| Health facility | 829 (78.4) |
| Out-reach | 228 (21.6) |
| Results of last HIV test[#], (n (%) | |
| Negative | 1233 (94.4) |
| Positive | 9 (0.7) |
| Didn't wait for results | 64 (4.9) |
| Disclosed HIV-status (Yes)[#], n (%) | 868 (66.5) |

* HIV risky behavior is a composite variable of having multiple sexual partners in last 6 months, inconsistent condom use with sexual partner(s), engaging in transactional sex and use of alcohol or drugs

[#] Among those who had ever-tested in their lifetime (N = 1306)

[+] Among those who have ever tested at a community service point (N = 1057)

*"My family members I told them, I told my children and my wives. . .I went ahead and told them that I am infected and they encouraged me to take HIV treatment. (Male, 32 years, HIV positive)*

Participants preferred to self-test in the morning when they have not eaten anything and saliva is pure.

*". . .test yourself in the morning. . . when you have just woken up because at that time you have not eaten anything, even the teeth will be undisturbed when saliva is still pure." (Male 34 years, HIV negative)*

## HIV testing and behavioral characteristics

Nearly half (49%) of the participants were circumcised and almost all engaged in at least one HIV risk behavior. Nineteen point eight percent had never tested for HIV in their lifetime while 43% had not tested in over a year. About 92.9% knew the HIV services available in their communities and 78.4% had ever tested at a community health facility. **Table 2** summarizes participants' HIV testing history and behavioral characteristics.

Participants reported that HIVST was more accessible because VHTs moved throughout the community and took testing services to men who could have been at risk of HIV infection but were unable to access the HIV services.

*"The person who influenced me most to use that method was the health worker, the way he came and explained to me the advantages which are in testing for this disease." (Male, 38 years, HIV negative)*

**Table 3. Uptake of oral HIVST among participants enrolled in the oral HIV self-testing cohort study in Mpigi district, 2018 (N = 1628).**

| Variable | Measure |
|---|---|
| Median days to use HIVST kit, (interquartile range) | 6 (1, 7) |
| Self-reported HIVST uptake (Yes), n (%; 95%CI) | 1603 (98.5; 97.13–99.18) |
| HIVST uptake verified by viewing the used kit (Yes), n (%; 95%CI) | 1564 (96.0; 94.09–97.40) |
| Negative | 1499 (95.8) |
| Positive | 63 (4.0) |
| Indeterminate | 2 (0.2) |
| Presenting self-reported HIVST uptake by HIV testing history*, n (%) | |
| Never tested | 317 (19.8) |
| Tested more than 12 months | 689 (43.0) |
| Tested in the last 12 months | 597 (37.2) |
| Location used for HIVST[+], n (%) | |
| Home | 617 (39.5) |
| Workplace | 723 (46.2) |
| Health facility | 98 (6.3) |
| Elsewhere | 126 (8.1) |
| Disclosed intention to use HIVST[+], n (%) | 1260 (80.6) |
| Person to whom intention to use HIVST was disclosed[a], n (%) | |
| Partner | 386 (30.6) |
| Friend | 842 (66.9) |
| Family member | 32 (2.5) |
| Needed help during HIVST (Yes)[+], n (%) | 326 (20.8) |
| Person asked for help during HIVST[b], n (%) | |
| Partner | 102 (31.3) |
| Friend | 202 (62.0) |
| Family member | 12 (3.7) |
| Village Health Team | 10 (3.0) |
| Overall disclosure of HIVST results, n (%) | 1183 (75.6) |
| Disclosure of HIVST results among those tested HIV-positive[c], n (%) | 35 (55.6) |
| Person to whom HIVST results were disclosed[d], n (%) | |
| Partner | 404 (34.2) |
| Friend | 740 (62.5) |
| Family member | 33 (2.8) |
| Village Health Team | 6 (0.5) |

* Among those who used HIVST (N = 1603)

[+] Among the proofed HIVST users (N = 1564)

[a] Among those who disclosed intention to use HIVST (N = 1260)

[b] Among those who needed support during HIVST (N = 326)

[c] Among those with HIV-positive HIVST results (N = 63)

[d] Among those who disclosed their HIVST results (N = 1183)

Nine (9) participants had HIV positive test results at their last time of testing. The HIV positive men reported that HIVST gave them a chance to verify the accuracy of the test, confirm their sero-status and commit to continue on ART.

> "...I knew that I had HIV because they had tested my blood before. I wanted to confirm the truth that comes from blood and saliva and I received the same answer... I went ahead with taking my drugs" (Male, 34 years, HIV-Positive)

## Uptake of oral HIVST

Self-reported uptake of HIVST was 98.5% and 4.0% of the participants tested HIV-positive. Forty six percent (46%) used HIVST from their workplaces; 80.6% disclosed their intention to self-test to significant others; 75.6% disclosed their HISVT results and 20.8% reported that they needed help during HIVST. **Table 3** shows results on the uptake of oral HIVST among study participants.

Over 95.8% of the participants tested HIV negative on HIVST. Men perceived HIVST as a confirmatory test. They reported that testing negative on HIVST and receiving confirmation from blood-based testing motivated them to embrace HIV prevention services.

> *"I was HIV negative after testing with saliva, I went to hospital and the results matched that I was negative so I said let me keep my life safe and from then I am using condoms." (Male, 26 years)*

Nearly 20% of the participants had never tested for HIV in their lifetime. They said that HIVST gave them an opportunity to test and disprove the suspicions they had about their health and lifestyles.

> *". . .I already had suspicions about my lifestyle; the partners I had. I wanted to test but I did not have the opportunity."(Male, 32 years)*

Participants self-tested from various locations that they found convenient including at workplaces (46.2%) and at home (39.5%). They said that HIVST was attractive because it offered flexibility of choosing where and when to perform the test.

**Table 4. Ranking of the oral HIVST process among participants enrolled in the oral HIV self-testing cohort study in Mpigi district, 2018 (N = 1628).**

| Variable | Measure |
| --- | --- |
| Had difficulties during HIVST (Yes), n (%) | 258 (16.5) |
| Difficulties found during HIVST*+, n (%) | |
| Following the pack insert guide | 198 (76.7) |
| Finding a private place to test | 177 (68.6) |
| Using the test materials | 64 (24.8) |
| Timing the test | 74 (28.7) |
| Reading results | 121 (46.9) |
| Interpreting results | 177 (68.6) |
| Found HIVST beneficial (Yes), n (%) | 1540 (98.5) |
| Benefits of HIVST+c, n (%) | |
| First to know own HIV-status | 1175 (76.3) |
| Privacy | 964 (62.6) |
| Short testing time | 803 (52.1) |
| Motivation to visit health facility | 448 (29.1) |
| Convenient testing | 769 (49.9) |
| Needed further support after HIVST (Yes), n (%) | 1066 (68.2) |

[b] Composite variable from Likert scale responses

*Among those who had difficulties during HIVST (N = 258)

+Multiple choice responses

[c] Among those who found HIVST beneficial (N = 1540)

*"Any moment I get my time whether I am from the garden, I can decide that in these 20 minutes let me test myself." (Male, 29 years, HIV negative)*

*". . .I pass it through my gum. I will put it there for a few minutes, to show me the truth and go back to my work without disturbing people that I will get transport, first line up and even come back without getting what I wanted in time". (Male, 35 years)*

## Participants' rating of the oral HIVST process

Sixteen point five percent reported difficulties during HIVST such as; following instructions on the kit insert (76.7%), finding a private place to test (68.6%) and interpreting results (68.6%). Participants' reasons for using HIVST included; being the first to know one's sero-status (76.3%), privacy (62.6%) and short testing time (52.1%), while 68.2% needed psychosocial support after HIVST. **Table 4** summarizes participants' ranking of the process of oral HIVST.

Most participants (98.5%) found HIVST beneficial because they learnt their HIV sero-status in private (62.6%) before anyone else (76.3%). They said HIVST gave them a chance to test and know their results in private unlike at health facilities where privacy is at times not assured.

*". . .you test yourself alone, which is a good thing because it is your secret. You know it privately than seeing you in public and somebody points at you that this one was told that he is sick at the hospital. . ." (Male, 29 years, HIV negative)*

*". . .test alone and know your status alone without anybody finding out. . .there are health workers who can't keep secrets. He can test you and then tells his colleague that big man you see is sick." (Male, 34 years, HIV negative)*

Men mentioned that HIVST helped to normalize HIV testing because they experienced less tension and no fear of pain.

*". . .it does not hurt my life that is the only difference I saw on it. . .to test myself and have no tension because of feeling pain." (Male, 29 years, HIV negative)*

Some participants (16.5%) reported finding difficulties during HIVST with; following instructions on the kit insert, opening and positioning of the developer liquid, timing of the test and interpretation of results. Men were concerned that oral HIVST was prone to error among amateur users, as they may fear to use the test, find the kit insert instructions unclear perhaps because of language barrier, or use the kit wrongly.

*". . .the person you would give it may fear to use it because sometimes the instructions can confuse. You may find a person putting the last thing first, not what you told him to do. Also, sometimes even in these paper instructions, the language; some of us we understand Luganda and you may not take me through it" (Male, 42 years, HIV positive)*

*". . .some people don't read the first step and proceed to the second. He just rushes even when he does not know how things go. . .may test himself and forgets that he tested and time exceeds. . ." (Male, 35 years, HIV negative)*

*". . .say the other liquid has poured, it is not going to give you correct results. . .It may show that you are positive yet you are negative. You remain in doubt and find that one who would start treatment has not, or the one who didn't need treatment has taken it which is also a problem." (Male, 32 years, HIV positive)*

More than half (68.2%) of the participants needed further support after HIVST. Men expressed need for counseling support to optimize benefits of HIVST and minimize negative consequences. They were concerned that without access to post-test counseling, HIVST may not prevent occurrence of self-harm, illicit behavior, and social conflict among users. They also reported of a likelihood of disclosure challenges after positive results since linkage to care is not guaranteed, and thought HIVST results could be used to decide on long term relationships.

*". . .a person can know, get his results, and may not want to go for counselling but remains there and does something wrong because of fear before going to get further assistance from the doctor and may commit suicide. . ." (Male, 32 years HIV negative)*

*". . . if he learns that he has it [HIV] he can say that I will not die alone let me go and give it to another person and I die with her. They need to be taught because somebody can tell you that we shall die ten of us because I also did not get it from a banana plantation." (Male, 35 years, HIV negative)*

*". . .I would first test with her using Oraquick and show her my results and if I find that she is negative then I can continue to marry her. If I find out that she is positive, I leave her and look for a fellow negative." (Male, 26 years, HIV negative)*

Men also said that they needed continuous sensitization about the available linkage services after HIVST and recommended using multi-media channels such as community radios, mobile vans and social media for sensitizing communities on HIVST.

*". . .help people who first test themselves alone, and first teach them like how you taught us because. . .if the government decides for everybody to test alone then do not sell those kits in shops such that people are first taught how to use that kit. . ." (Male, 35 years, HIV negative)*

## Discussion

We conducted a concurrent parallel mixed methods analysis to assess the uptake and percep-tions of VHT-delivered oral HIVST among 1628 men in Mpigi district in Central Uganda. We found a high HIVST uptake of 96% with 19.8% first-time testers, 4% testing HIV positive and 75.6% disclosed their HIVST results to sexual partners or significant others. Men perceived HIVST as a quick, flexible, convenient and private approach to HIV testing; allowing easy dis-closure of HIV test results and receiving social support, and an opportunity for knowing or re-confirming their HIV sero-status, and subsequent linkage or re-linkage to care and prevention services. Some men however, expressed concern that HIVST was prone to error and a few found difficulties in performing the HIVST. Some men also expressed the need for post-test counselling support and additional information on linkage to services after HIVST. Our find-ings strengthen evidence that VHT-delivered HIVST is a promising and feasible approach for reaching men with HIV testing services including receiving information about HIV preven-tion and treatment.

Our findings showed a high uptake of VHT-delivered HIVST among men. These results are similar to prior work done elsewhere in SSA, where community models increased HIV testing uptake, the proportion of first-testers and untreated HIV identification yield as com-pared to conventional approaches [23, 45, 48–51]. Our study population was mainly young people who were receptive to new health care technologies [52, 53] and could easily learn how to use HIVST to test at their own convenience [34, 44, 45]. Young people are a priority sub-

population for closing the gap to end HIV by 2030 [4, 21, 54]. In the 2022 UNAIDS (The Joint United Nations Program on HIV/AIDS) HIV update, 28% of new HIV infections in Eastern and Southern Africa were among male young people aged 15–49 years, and 53% among their female counterparts [1]. Although these statistics show a higher risk among women than men, access and utilization of essential HIV services is still lower among men necessitating more innovative approaches to ensure service use among men living with or at risk of HIV, to improve their health outcomes, and prevent HIV transmission to their sexual partners [1]. The high uptake is also explained by the mostly positive perceptions of HIVST discussed below. Our results therefore contribute to evidence recommending community-based models of HIVST as a potential approach for extending HIV testing services to men and sub-groups often missed by conventional HIV service models.

We found 4% (63) HIV positivity among men who self-tested (54 new diagnoses and 9 untreated but known HIV positive). These results show that the potential of community-based HIVST in reaching the remaining undiagnosed minority and those at high risk of HIV infection. They also justify the need for innovative testing approaches including combined community and facility-based strategies to close the remaining 5% gap in achieving the first target of the UNAIDS HIV goals [1, 55]. In addition, these results agree with the latest national statistics on HIV among adults in Uganda which reported a prevalence of 4.3% among men and 7.2% in women [5]. Our study population was predominant young people, so our results affirm the observed 53% reduction in new HIV infections among young people aged 15–24 years nationally [56]. Although HIV prevalence remains higher among young women compared to the men of the same age group [5, 56], the number of men accessing and utilizing HIV testing services is lower [4, 31, 56]. Our work therefore, emphasizes the need for innovative gender-friendly approaches to reach men with HIV testing services to improve their HIV-positive status awareness.

Our results showed a high disclosure of HIVST results to sexual partners or significant others. However, disclosure to sexual partners (34.2%) was lower than to friends (62.5%) and even lower (55.6%) among those that tested HIV positive. Disclosure is an essential part of the HIV testing and treatment cascades as it facilitates relief and sero-status acceptance, empowerment, access, adherence and retention in HIV care, and enhances physical and psychosocial support from loved ones and partner(s) [57–62]. In the context of HIVST, there is still need for strategies to increase disclosure [1]. In the ATLAS project in Mali, health workers hesitated to distribute HIVST kits to PLHIV who had not disclosed their sero-status to sexual partners, or when uncertain of how to support status disclosure for known PLHIV [63]. They rather recommended optimizing engagement of VHTs and social workers in HIVST to avail more time and social comfort for users to disclose their test results [63]. In our earlier research, and elsewhere in SSA, disclosure of HIVST results had a positive influence on seeking of facility-based services including HIV confirmatory testing, anti-retroviral therapy (ART) initiation and care [23, 44, 49], but access to care could be disrupted in case of discordancy [64]. Our study utilized VHTs to deliver HIVST to reach men in the community, hence contributes to evidence exploring approaches for improving HIVST uptake and sero-status disclosure among priority populations like men. Nevertheless, there is need to optimize disclosure to sexual partners or those tested HIV positive in order to maximize benefits of HIV testing, for accessing appropriate prevention and care services.

Our study found that men mostly had perceived HIVST such as being a flexible and convenient testing approach for learning or being up-to-date with one's HIV status. These findings are consistent with earlier research demonstrating the potential of HIVST in breaking boundaries of time and place to provide quick, accurate, private and convenient testing to the underserved, hard-to-reach and priority populations such as men [30]. A systematic review of

qualitative evidence on factors enabling and deterring uptake of HIV self-testing in Africa [27], reported several perceived facilitators including; autonomy and self-empowerment, privacy, confidentiality, convenience, opportunity to test, including couples HIV testing and ease of use [27]. They also reported numerous barriers including; cost of HIVST kits, perceived unreliability of HIVST results, low literacy, fear and anxiety of HIV positive results, potential of psychological and social harms, potential violation of human rights, concerns over- face-to-face counseling, linkage to care, and regulatory and quality assurance systems, and quality of self-test kits [27]. Since HIVST was a relatively new testing approach in Uganda [31], some men had reservations similar to those reported by Njau et al [27] such as how to use the kit, resolving of potential occurrences of self or social harm especially in unsupervised HIVST, and access to post-test support following HIVST. Although these concerns are reported to deter implementation of HIVST [30, 63, 65–67], our study did not register any negative occurrences linked to HIVST. Regarding the need for support after HIVST, our research showed that VHT-delivered HIVST accompanied by information leaflets had a positive influence on the utilization of the surrounding facility-based HIV prevention and treatment services [44]. Our findings reinforce the advantages of HIVST in enabling HIV testing among priority populations and highlight areas of caution during implementation and future scale-up of HIVST. Achieving population level coverage of men's HIV testing may therefore require continuous sensitization of target populations on the novel approaches like HIVST and provision of counselling to dispel negative perceptions of HIVST [4, 30, 51].

Our study has several strengths. First, we utilized VHTs to deliver HIVST. VHTs comprise part of the Ugandan health system, supporting provision of essential health services to communities. VHTs have the potential to distribute HIVST alongside other services to enable easier access to and utilization of HIV testing services for men and other missed minority groups. A VHT-led model of HIVST is sustainable, replicable and can be scaled-up by programs for rolling-out HIVST across the country. Second, we conducted our study among men in a generalized HIV epidemic so our results can be applied in similar settings in Uganda and other resource-limited settings in SSA. Third, we used a robust mixed methods study design to collect data and report our findings and so minimized the potential for information bias arising from the individual study approaches by complementing the quantitative results with qualitative findings.

Some of the limitations in our study were that: 1) we measured HIVST uptake by self-reports and verified it by viewing the used kit. However, it is possible that some responses were false or that individuals who received the HIVST kits were not the ones who eventually self-tested. We believe the false reports were minimal since there was no statistically significant difference between the HIVST self-reports and verified HIVST results. Self-reports in health care have been shown to have a substantial degree of accuracy [68]. 2) Our eligibility criteria was restricted to participants who had access to a functional mobile phone so it possible that we may have had selection bias. However, we believe this was minimal since participants who did not own phones had access to phones of others in their households and the VHTs readily shared their contacts as alternative for communication when following up participants in their communities. 3) Our participants were only cis–gender men identifying with the male gender, so we may have missed men with varied sexual orientations hence our findings may not be generalizable to men among the gender minorities. Nonetheless, selection bias arising from this may be minimal since men in the community predominantly identified as male. 4) The qualitative results presented in this paper, were from only men who used HIVST so do not give insights on HIVST non-use which may guide future efforts in reaching the remaining few untested men missed by our HIVST approach.

## Conclusion

In conclusion, our VHT-led delivery model of HIVST resulted in a high uptake of HIV testing services among men living in a semi-urban setting of Uganda. In addition, the men reported mostly positive but also some negative perceptions that would require counseling support during roll out of HIVST programs. Given that this was a community-based testing, the HIV positive yield was high and included identification of men with untreated HIV. Thus, our model has potential for identifying HIV positive men who do not test during conventional testing programs and can easily be scaled-up using the existing network of VHT in Uganda or similar community health workers in other countries. Future implementation research could explore approaches to optimize disclosure of HIVST results especially among those tested HIV positive.

## Acknowledgments

We greatly appreciate the support rendered by the Ministry of Health directorate in Mpigi district and all the local areas authorities in the study villages. We also extensively appreciate all the participants in this study and the Makerere University Implementation Science program for the opportunity contribute to the knowledge in this field.

## Author Contributions

**Conceptualization:** Joanita Nangendo, Anne R. Katahoire, Charles A. Karamagi, Gloria O. Obeng-Amoako, Joan N. Kalyango, Rhoda K. Wanyenze, Moses R. Kamya.

**Data curation:** Joanita Nangendo, Gloria O. Obeng-Amoako, Mercy Muwema, Jaffer Okiring, Fred C. Semitala, Joan N. Kalyango.

**Formal analysis:** Joanita Nangendo, Jaffer Okiring, Joan N. Kalyango, Moses R. Kamya.

**Funding acquisition:** Joanita Nangendo, Fred C. Semitala, Moses R. Kamya.

**Investigation:** Joanita Nangendo, Anne R. Katahoire, Charles A. Karamagi, Gloria O. Obeng-Amoako, Mercy Muwema, Jane Kabami, Moses R. Kamya.

**Methodology:** Joanita Nangendo, Anne R. Katahoire, Charles A. Karamagi, Gloria O. Obeng-Amoako, Mercy Muwema, Jaffer Okiring, Jane Kabami, Joan N. Kalyango, Rhoda K. Wanyenze, Moses R. Kamya.

**Project administration:** Joanita Nangendo, Gloria O. Obeng-Amoako, Mercy Muwema, Jane Kabami.

**Supervision:** Anne R. Katahoire, Charles A. Karamagi, Fred C. Semitala, Joan N. Kalyango, Rhoda K. Wanyenze, Moses R. Kamya.

**Validation:** Mercy Muwema, Jaffer Okiring, Jane Kabami, Fred C. Semitala, Joan N. Kalyango, Rhoda K. Wanyenze.

**Writing – original draft:** Joanita Nangendo, Anne R. Katahoire, Moses R. Kamya.

**Writing – review & editing:** Joanita Nangendo, Anne R. Katahoire, Charles A. Karamagi, Gloria O. Obeng-Amoako, Mercy Muwema, Jaffer Okiring, Jane Kabami, Fred C. Semitala, Joan N. Kalyango, Rhoda K. Wanyenze, Moses R. Kamya.

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
