## [Decision Letter · Decision Letter 0]

25 Apr 2023

PGPH-D-23-00366

Uptake and perceptions of oral HIV self-testing delivered by Village Health Teams among men in Central Uganda: A concurrent parallel mixed methods analysis

Dear Dr. Nangendo,

Thank you for submitting your manuscript to PLOS Global Public Health. After careful consideration, we feel that it has merit but does not fully meet PLOS Global Public Health’s publication criteria as it currently stands. Therefore, we invite you to submit a revised version of the manuscript that addresses the points raised during the review process.

We look forward to receiving your revised manuscript.

Kind regards,

Ama Pokuaa Fenny, Ph.D

Academic Editor

Journal Requirements:

Additional Editor Comments (if provided):

Reviewers' comments:

Reviewer's Responses to Questions

**Comments to the Author**

1. Does this manuscript meet PLOS Global Public Health’s publication criteria? Is the manuscript technically sound, and do the data support the conclusions? The manuscript must describe methodologically and ethically rigorous research with conclusions that are appropriately drawn based on the data presented.

Reviewer #1: Yes

Reviewer #2: Yes

2. Has the statistical analysis been performed appropriately and rigorously?

Reviewer #1: N/A

Reviewer #2: Yes

3. Have the authors made all data underlying the findings in their manuscript fully available (please refer to the Data Availability Statement at the start of the manuscript PDF file)?

Reviewer #1: Yes

Reviewer #2: No

4. Is the manuscript presented in an intelligible fashion and written in standard English?

Reviewer #1: Yes

Reviewer #2: Yes

5. Review Comments to the Author

Reviewer #1: Thank you for inviting me to review "Uptake and perceptions of oral HIV self-testing delivered by Village Health Teams among men in Central Uganda: A concurrent parallel mixed methods analysis"

Comments # 1: In Abstract

Methods

Line number 35: What types of analysis were used for quantitative data? Please mention it. In this section, the authors only highlighted qualitative data analysis.

Results

Who were your significant others? Was there any association? Please highlight these.

Comments # 2: Introduction

The author should include some references that have summarized HIV status awareness across the globe. The author also ought to include some studies that have been conducted not only in developing countries but also in developed countries to support this manuscript. The introduction part highlighted only Uganda and Sub-Saharan African-related studies

Comments # 3: Methods

Study design and setting

What were the reasons for conducting this study only for men except for women? Was any previous study available among women in this region?

How many Village Health Teams (VHTs) were directly involved to conduct this survey? Please mention.

Study participants:

Why the study was selected ≥ 15 years age of the study participants? ≤ 15 years age of the participants were safe from HIV or the prevalence rate was too little in this region or no statistics were available?

Line number 107-108: This is unnecessary. Please omit this line.

Sampling and sample size estimation

Please highlights the sample size estimation formula.

Line number 117: How many participants were included in the qualitative study? Please clear it

Study procedures

Which type of questionnaires were used in the in-depth interview? Was it structured/unstructured/ semi-structured?

Data analysis

The data were only generalized by percentage. Please revise the analysis and perform a higher level of analysis to find out the significant association between different variables.

Comments # 4: Results

Line number 180: Please use the numerical value 25 instead of twenty-five participants

Table 1: Please correct the spelling of Moslem

Table 2: Please revise the variables tested =>12 months ago

Line number 239: Please use the numerical value 9 instead of nine participants

Line number 247: Please use the numerical value 46% in place of forty-six percent

Table 3: Please mention the AOR in the rows of Self-reported HIVST uptake (Yes), n (%; 95% CI), and HIVST uptake verified by viewing the used kit (Yes), n (%; 95%CI)

Comments # 5: Discussions

The discussion sections should be revised.

Line number 400: Please highlight the elaboration of PWH before using the abbreviations.

Comments # 5: References

Some references were used in this manuscript which is more than 10 years. Please updated the references including 20, 30, 37, 41, 49, 58, and 66.

Overall comments:

There are some grammatical errors as well as improving the English language. It is noticed that the structure of many sentences is misleading. Please revise the manuscript correctly, avoiding grammatical errors.

Reviewer #2: It's a pertinent study to understand the uptake of HIV self-testing kits and the perception of this service delivered through a local team like village health team in Mpigi district in Uganda. It has the potential to inform implementation of programmes that are community based for better uptake of HIV testing. The research question, objectives, methods are all coherent and are explained in depth. Results are well structured and laid out. A few minor clarifications to be given by the authors in the methods -

1. Rationale behind choosing 10 days as duration after which follow up was done to see if the test was utilised can be explained.

2. Whether experiences of the village health teams was considered in order to comment of feasibility of such a programme including the challenges? If not maybe mention this as a limitation or further area to be explored before scaling this up.

3. Marital status was asked to all men - were only heterosexual men included in the study? Was sexual orientation asked about? if so please specify if there were any implicit assumptions related to this and state under methods/limitations accordingly.

4. There is a chance that the choice to purposely interview people who utilised the test kit and those who volunteered among them to share their experience, would have influenced the perceptions shared instead of exploring both those who utilised and did not utilise the kit. Which might have given a better understanding. This is something to consider for the limitation section or explain the rationale behind this criteria.

6. PLOS authors have the option to publish the peer review history of their article (what does this mean?). If published, this will include your full peer review and any attached files.

**Do you want your identity to be public for this peer review?** For information about this choice, including consent withdrawal, please see our Privacy Policy.

Reviewer #1: **Yes: **Md. Zohurul Islam

Reviewer #2: No

---

## [Editor Report · Decision Letter 1]

16 May 2023

Uptake and perceptions of oral HIV self-testing delivered by Village Health Teams among men in Central Uganda: A concurrent parallel mixed methods analysis

PGPH-D-23-00366R1

Dear Ms Nangendo,

We are pleased to inform you that your manuscript 'Uptake and perceptions of oral HIV self-testing delivered by Village Health Teams among men in Central Uganda: A concurrent parallel mixed methods analysis' has been provisionally accepted for publication in PLOS Global Public Health.

Best regards,

Ama Pokuaa Fenny, Ph.D

Academic Editor